# Non-Coding RNA in Tumor Cells and Tumor-Associated Myeloid Cells—Function and Therapeutic Potential

**DOI:** 10.3390/ijms25137275

**Published:** 2024-07-02

**Authors:** Amanda Katharina Binder, Franziska Bremm, Jan Dörrie, Niels Schaft

**Affiliations:** 1Department of Dermatology, Universitätsklinikum Erlangen, Friedrich-Alexander-Universität Erlangen-Nürnberg, 91054 Erlangen, Germany; amanda.binder@uk-erlangen.de (A.K.B.); franziska.bremm@uk-erlangen.de (F.B.); jan.doerrie@uk-erlangen.de (J.D.); 2Comprehensive Cancer Center Erlangen European Metropolitan Area of Nuremberg (CCC ER-EMN), 91054 Erlangen, Germany; 3Deutsches Zentrum Immuntherapie (DZI), 91054 Erlangen, Germany; 4Bavarian Cancer Research Center (BZKF), 91054 Erlangen, Germany

**Keywords:** non-coding RNA, cancer therapy, tumor-micro-environment, tumor-associated macrophages, tumor-associated dendritic cells, myeloid-derived suppressor cells

## Abstract

The RNA world is wide, and besides mRNA, there is a variety of other RNA types, such as non-coding (nc)RNAs, which harbor various intracellular regulatory functions. This review focuses on small interfering (si)RNA and micro (mi)RNA, which form a complex network regulating mRNA translation and, consequently, gene expression. In fact, these RNAs are critically involved in the function and phenotype of all cells in the human body, including malignant cells. In cancer, the two main targets for therapy are dysregulated cancer cells and dysfunctional immune cells. To exploit the potential of mi- or siRNA therapeutics in cancer therapy, a profound understanding of the regulatory mechanisms of RNAs and following targeted intervention is needed to re-program cancer cells and immune cell functions in vivo. The first part focuses on the function of less well-known RNAs, including siRNA and miRNA, and presents RNA-based technologies. In the second part, the therapeutic potential of these technologies in treating cancer is discussed, with particular attention on manipulating tumor-associated immune cells, especially tumor-associated myeloid cells.

## 1. Introduction

### 1.1. RNA Is More Than a Blueprint for Proteins

Proteins are encoded by DNA, yet DNA is not directly translated into polypeptides. First, the information is transcribed into RNA, which functions as a vehicle to transport the information from the nucleus to the production site. The importance of RNA culminates in the RNA world hypothesis, which suggests that RNA precedes DNA as genetic material and proteins as biological enzymes, thus forming the archetype molecule of life [1]. In recent years, the therapeutical application of (m)RNA became feasible since biochemical modifications and new shuttling systems allowed higher stability of the RNA, lower immunogenicity, and a protected delivery within the human body. The field of mRNA-based immunization became very popular due to the coronavirus pandemic.

However, besides mRNA and its functions, several other RNA types have been identified in plants, bacteria, and animals [2]. RNA-interactome studies revealed that these RNAs are able to interact with other RNAs, DNA, proteins, and peptides and exert crucial functions in the regulation of replication, transcription, and translation [3]. In addition to these ubiquitous functions, the non-coding RNAs are central in multiple cellular defense systems against viruses (e.g., double-stranded RNA of viral origin) and mobile genetic elements (e.g., transposons, plasmids, and phages) [3,4]. Hence, many functional principles involving non-coding (nc)RNA were first observed and characterized in the context of pathogen defense but can be exploited for new therapeutic approaches. A part of these defense systems utilizes small guide RNAs that recognize aberrant transcripts and mediate their inactivation [5]. This mechanism of action is termed RNA interference (RNAi) and subsumes several molecular mechanisms that use small guide RNA fragments to target specific nucleic acid sequences. The recognition of the target is mediated via complementary base pairing, which requires a reverse complementary target sequence and is therefore referred to as antisense regulation [5,6].

### 1.2. A Brief History of RNA-Based Regulation

Historically, gene regulation by RNAs was already proposed by Francois Jacob and Jacques Monod in 1961 [7] but could not be proven experimentally at this time. In the 1980s, antisense RNA control was thoroughly studied in bacteria and phages [8]. In the 1990s, a key experiment with eukaryotic RNAi was conducted by Fire and Mellow in *Caenorhabditis elegans*, showing that artificially introduced short antisense RNA was able to down-regulate the expression of developmental genes on the translational level by binding to the mRNA of the target gene [9,10,11]. This work won the Nobel prize in 2006, and RNAi was judged to be “a fundamental mechanism for controlling the flow of genetic information in cells” [12].

In 2020, another system that utilizes short non-coding RNA in a similar fashion to RNAi won the Nobel prize: the nucleic acid-guided defense systems of the bacterial adaptive immune system called CRISPR/Cas [13,14]. In prokaryotes, the clusters of regulatory interspersed short palindromic repeats (CRISPR) are elements in the original bacterial DNA, and the sequences between the repeats represent genetic information from previous invaders that integrated into the bacterial DNA. Upon transcription of the CRISPR DNA elements, CRISPR-RNA is formed, which is complementary to the original pathogenic genetic material, and is thereby able to recognize the pathogens with this sequence upon repeated intrusion. Since this system is a defense mechanism, the binding of the CRISPR-RNA initiates the cleavage of the foreign genetic material by endonucleases of the Cas family [15,16,17,18]. 

Evolutionarily, the adaptive immunity in eukaryotes was mainly taken over by protein-based recognition molecules, while RNAi still remains a central gene regulatory mechanism in eukaryotes [5,19]. In the following section, a brief overview of eukaryotic RNA forms and their functions is given, with a special focus on endogenous sequence-specific gene silencing on the post-transcriptional level. 

## 2. RNAs in Eukaryotes 

Endogenously expressed RNAs in eukaryotes can be divided into different subclasses based on their biogenesis, general characteristics, and functions. The most prominent subdivision distinguishes protein-coding RNAs (mRNAs) and non-protein coding RNAs (ncRNAs) [20,21].

mRNAs with sizes up to several kilobases (kb) are structurally well-defined by being composed of the protein-coding region for translation as well as 3′ and 5′ untranslated regions (UTRs). First, a 7-methylguanosin is added as a cap at the 5′UTR that protects the mRNA from degradation. At the end of the 3′UTR, the mRNA is polyadenylated, thus forming a so-called Poly(A) tail that enhances the stability and is involved in the transport of the mRNA [22]. mRNAs are processed in the nucleus, after which the matured mRNAs are shuttled into the cytoplasm. Finally, the translation into a protein occurs at ribosomes [23]. 

Ribosomal RNA (rRNA) and transfer RNA (tRNA) are functional RNAs that are highly abundant and form their own subclasses. Both are constitutively expressed and harbor housekeeping functions involved in general protein synthesis at the ribosomes. Although they do not encode proteins, they are not considered part of the group of ncRNAs. Their biogenesis and functions are reviewed extensively elsewhere [24,25,26]. 

In contrast to mRNA, rRNA, and tRNA, ncRNAs comprise a diverse group with less well-defined subgroups and continuously new discoveries of RNA types. However, the ncRNAs can be broadly grouped by their size [27,28]. The long non-coding RNAs (lncRNAs) are, per definition, larger than 200 nucleotides (nt), often, but not always, capped and polyadenylated, and are present in the nucleus and the cytoplasm [29,30,31]. This group includes, for example, enhancer RNAs (eRNA) which mediate mRNA transcription [32,33]. Their biological function is not yet fully agreed upon and is extensively reviewed elsewhere [34,35]. Another example of this size group is circular RNAs (circRNA), which are a product of non-canonical splicing and have different regulatory functions, including the regulation of other RNAs and proteins [36]. 

The second group is small ncRNAs with sizes smaller than 200 nt. One representative of this group is P-element-induced wimpy testis (PIWI)-interacting RNAs (piRNA). Especially in germ-line cells, this functional complex of protein and RNA protects the genome from transposons by chromatin modifications that repress transcription [37,38]. Other small ncRNAs, such as small nuclear RNA (snRNA), act in the cell nucleus or nucleoli that are involved in the splicing of pre-mRNA [39,40], and small nucleolar RNA (snoRNA) that mediate the processing of other RNAs, e.g., of rRNA [41]. Lastly, micro-RNAs (miRNA) and small-interfering RNAs (siRNA), both 19–25 nt long, act on the translation of mRNA and are therefore the two main RNA types involved in post-transcriptional gene silencing [19,42,43,44]. Since they play a prominent role in gene regulation and their regulatory function holds great potential in therapeutic application, both are discussed in detail below.

### 2.1. Biogenesis of Micro-RNAs and Small-Interfering RNAs 

The discrimination between miRNA and siRNA is difficult since both have similar mechanisms of action. So far, the main distinction is made by their origin and biogenesis. miRNAs, as endogenous regulators of mRNA translation, are always genome-derived, originating from introns of protein-coding genome sequences, from introns or exons of ncRNA genome sequences, or from independent transcriptional units [45]. The biogenesis of miRNAs via the canonical pathway starts with the transcription of the above-mentioned gene loci by RNA polymerase II/III, producing the so-called primary miRNA (pri-miRNA) present in the nucleus [46]. Structurally, this pri-miRNA forms a hairpin-shaped RNA construct with complementary-bound double-stranded RNA (dsRNA) parts and an apical loop [47]. Still in the nucleus, the pri-miRNA is cleaved into one or more precursor miRNA (pre-miRNA) by the microprocessor complex composed of the enzymes Drosha and the co-factor Pasha [48,49] (Figure 1). Thereafter, the pre-miRNA is shuttled into the cytoplasm via Exportin 5 [50]. In the cytoplasm, the double-stranded part of the pre-miRNA binds to a ribonuclease III enzyme called Dicer [51], which, in turn, cleaves the pre-miRNA into a miRNA duplex composed of the guide miRNA single-strand and a complementary bound passenger strand termed miRNA* [52,53]. The non-canonical pathway utilizes different combinations of enzymes apart from Drosha and Dicer, which is reviewed by Ha et al. [46].

In contrast to miRNAs, siRNAs are primarily of exogenous origin, deriving from double-stranded RNA transcripts of viruses or transgenes. Those can originate from exogenous sequences already integrated into the host genome, termed endo-siRNA, or from ex-genomic sources (exo-siRNA), including pathogens or artificial exogenously transferred dsRNA in experimental settings [45,54,55,56]. For the endo-siRNA, the resulting RNA transcripts contain complementary double-stranded sequences and do not undergo extensive processing steps in the nucleus but are rapidly shuttled into the cytoplasm [11,45,57]. For the exo-genomic siRNA, the dsRNA strands are solely present in the cytoplasm. Without regard to their origin, these dsRNAs are recognized by Dicer and enter the RNAi pathway [45,58]. The resulting 21–25 nt long RNA molecules are termed siRNA duplexes. Similar to the miRNA, a siRNA duplex consists of a guide strand and a passenger strand. It should be mentioned that the term siRNA often refers only to synthetic components since it is prominently used in experimental settings, although, as explained above, this must not always be the case.

**Figure 1 ijms-25-07275-f001:**
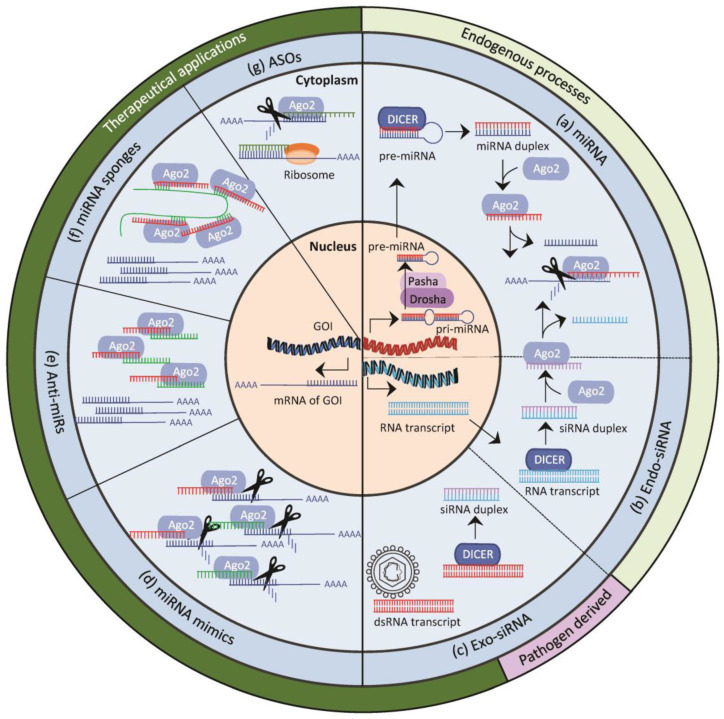
Overview of the mechanisms of action of naturally occurring RNAi pathways and therapeutic approaches intervening with the endogenous RNA interference (RNAi) pathway. (**a**) micro-RNA (miRNA) pathway. In the nucleus, the primary miRNA (pri-miRNA) is transcribed and processed by the enzyme Drosha and its co-factor Pasha into precursor miRNA (pre-miRNA). The pre-miRNA is shuttled into the cytoplasm and recognized by Dicer as double-stranded RNA (dsRNA) transcript. Dicer cleaves the pre-miRNA into the short miRNA duplex, which is recognized by Argonaute 2 (Ago2), which directs the binding to the mRNA targets and mediates the cleavage. (**b**) endo-siRNA. The dsRNA transcript originates from the genome in the nucleus and is shuttled into the cytoplasm, where it is recognized by Dicer. Dicer cleaves the long RNA transcript into the short small-interfering RNA (siRNA) duplex, which enters the RNAi pathways as described for miRNAs. (**c**) exo-siRNA pathway. The dsRNA transcript can be pathogen-derived (e.g., viruses) or therapeutically introduced into the cells. The dsRNA remains in the cytoplasm and is directly recognized by Dicer. (**d**) miRNA mimics. The miRNA mimics with the same sequence as the endogenous miRNA are recognized by Ago2 and direct the cleavage of the referring target mRNA. (**e**) Anti-miRs complementarily bind to the guide strand of endogenous miRNA, thus preventing binding to the target mRNA and subsequent cleavage. (**f**) miRNA sponges catch miRNAs by binding the seed region, also preventing the cleavage of the target mRNA. (**g**) ASOs. Anti-sense oligos (ASOs) have different mechanisms of action. In the cytoplasm, they can direct the cleavage of target mRNA via RNAi but can also inhibit the mRNA translation via steric hindrance at the ribosomes. The alternative mechanisms of action of ASOs within the nucleus are depicted in the following Figure in Section 3.1.

### 2.2. Micro-RNA and Small-Interfering RNA-Mediated Gene Silencing 

miRNAs and siRNAs suppress gene expression via the RNAi pathway, which requires an effector protein complex termed RNA-induced silencing complex (RISC) [59,60] (Figure 1). In the process, the short miRNA or siRNA duplex binds to the RNase H enzyme called Argonaute 2 (Ago2), the duplex gets unwound, and only the guide strand remains at the Ago protein, forming a miRISC or siRISC, while the respective passenger strand is degraded. In both cases, the retained miRNA or siRNA single-strands act as specific guide elements of the complex. Structurally, the recognition of the target sequence is mediated by the seed region, a heptameric (7 nt) sequence within the miRNA or siRNA [61,62,63]. The actual binding occurs via Watson–Crick base pairing to complementary mRNA sequences, mostly located in the 3′UTR of target mRNAs [52]. Upon a high level of complementarity, the target mRNA is cleaved directly [64,65,66], while loose binding, resulting from a partial mismatch, only leads to repression of mRNA translation [67,68,69,70]. miRNAs are able to act via both binding modalities, while siRNAs carry out their function upon complete complementary binding only [70,71]. 

### 2.3. Therapeutic Potential of Micro-RNA and Small-Interfering RNA

As outlined above, miRNAs and siRNAs are critical regulators of eukaryotic gene expression, thereby directing developmental processes, cell functions, and physiological processes [72]. In line with that, approximately 60% of the human protein-coding genes contain conserved miRNA-binding sites, implying that those miRNAs can control the expression of the majority of protein-coding genes [73,74]. Up to now, nearly 2700 identified human miRNAs are registered in the open data base miRBase (accessed March 2024). Moreover, dysregulated miRNAs or whole miRNA families are associated with the pathogenesis of human diseases, including CNS disorders, cardiovascular and metabolic diseases, and cancer [72]. Consequently, therapeutics that intervene in the miRNA regulation network or exploit siRNA functions are a powerful tool in several diseases. 

## 3. RNA Constructs in Therapeutic Approaches 

Upon using RNAs as therapeutic agents in vivo, three major challenges arise: (i) poor in vivo stability of RNA, (ii) delivery into the right tissue and through the cell membrane, and (iii) off-tissue effects caused by the binding to the target mRNA or to mRNAs with similar sequences in other cells and tissues. However, with ongoing improvement in all three fields, RNAs as posttranscriptional regulators are the subject of basic and clinical research [75,76,77].

### 3.1. Anti-Sense Oligonucleotides

Anti-sense-oligonucleotides (ASOs), also known as anti-sense therapy, were the first therapeutic approach exploiting complementary nucleic acid strands to interfere with gene expression [78]. ASOs are 12–25 nucleotides long oligomers of single-stranded DNA or RNA [79] (Figure 1). These chemically synthesized oligomers are designed to bind a specific nucleic acid sequence, such as mRNA or its nuclear precursor (pre-mRNA), via complementary base-pairing [80,81]. DNA-based ASOs are able to enter the nucleus of cells and form a DNA-RNA heteroduplex, leading to RNA degradation by activating RNase H, thereby preventing the expression of the respective protein [82] (Figure 2). Further, ASOs can induce altered splicing of pre-mRNA in the nucleus by mediating exon skipping or exon retention [83,84] (Figure 2). In the cytoplasm, DNA- and RNA-based ASOs act via steric hindrance, thus inhibiting translation of the mRNA at the ribosomes [85], as well as inducing mRNA degradation via RNAi [84] (Figure 1). However, a remaining problem of antisense therapy with single-stranded ASOs is chemical instability, low potency, and strong side effects due to the unwanted repression of mRNAs with similar sequences [79,86].

### 3.2. Small-Interfering RNA

Synthetic siRNA forms are a more specific and potent alternative to ASOs. The siRNA is introduced as a more stable double strand and exploits the cells’ intrinsic RNAi pathway, resulting in a sequence-specific inhibition of mRNA expression. Since siRNAs need to bind completely complementarily to exert mRNA degradation, the off-target effects are strongly diminished compared to ASOs [78] (Figure 1).

### 3.3. Micro-RNA 

The above-described anti-sense therapy can be exploited within the complex miRNA network. By utilizing or targeting miRNAs therapeutically, the regulation of the mRNA expression can be altered in several manners. Firstly, miRNA replacement therapy can be used to enhance endogenous miRNA activity by the addition of synthetic miRNAs that mimic endogenous miRNAs (miRNA mimics). Consequently, the gene expression can be further inhibited, leading to a more effective knockdown in the expression of the affected protein. In contrast to that, synthetic single-strand RNAs acting as miRNA antagonists (termed antagomirs or anti-miRs) can be introduced. These bind to the endogenous mature miRNA and inhibit the miRNA-mediated repression of gene expression (Figure 1). Hence, gene expression can be restored [72,87].

The latter approach is exceeded with miRNA sponges [88]. The sponge RNA can be delivered via a transgene encoded by a viral vector. These transcripts are expressed from strong promoters and contain multiple complementary binding sites only to the heptameric seed regions of the miRNAs of interest (see Section 2.2). Since seed regions are conserved, whole miRNA seed region families can be functionally blocked; thereby, the gene expression of several genes can be rescued [88,89] (Figure 1).

### 3.4. Micro-RNAs and Small-Interfering RNAs in Clinical Use

In 2018, the FDA’s approval of the first siRNA drug called Patisiran marked a new era of RNAi therapeutics. Patisiran was approved for treating the rare and lethal genetic disease hereditary transthyretin (TTR) amyloidosis. From a mutated allele, defective, mis-folded TTR proteins form, causing severe polyneuropathy and cardiac symptoms. Patisiran binds to the mutated mRNA, causing its degradation and preventing the formation of defective proteins [90,91]. Clinical data showed that seventy percent of the patients receiving Patisiran reached a stable or improved neuropathological stage compared to the placebo group, while side effects were balanced between the groups. However, it should be mentioned that atrioventricular heart blockade as a severe side effect in the Patisiran group was observed [92]. For the treatment of genetic diseases, the FDA approved the three siRNA active agents, Givosiran, Lumasiran, and Inclisiran, in 2019, 2020, and 2021, respectively [93]. Givosiran is a medication for acute hepatic porphyria [94]. Lumasiran was developed as a treatment for primary hyperoxaluria type 1 [95]. Inclisiran was approved for the treatment of familial hypercholesterolemia and reduces the low-density lipoprotein cholesterol levels [96].

Next to these rare genetic diseases, several new siRNA and miRNA therapeutics are in clinical trials targeting a variety of diseases, including cancer. Table 1 subsumes all miRNA and siRNA therapeutics for cancer therapy recently tested in clinical trials (information from ClinicalTrials.gov, [97]). Of note, all approaches listed below have in common the fact that cancer cells are the primary target of miRNAs and siRNAs. Several trials use siRNA to target a tumorigenic driver directly. The drug siG12D-LODER™ targets the mutated form of Kirsten rat sarcoma virus protein (KRAS) called KRAS^D12G^, which is a hallmark of several cancer types, including pancreatic cancer. KRAS naturally induces survival and proliferation of cells, which is exploited by cancer cells with KRAS mutations [98]. The therapeutic inhibition of KRAS expression abrogated this cancer-promoting effect. In a phase I study, it has been shown that the siG12D-LODER™ treatment led to stable disease or partial response in patients with locally advanced pancreatic cancer [99], which then proceeded to a phase II study with unknown status. The clinical phase I trial NCT03608631 also targets KRAS^D12G^ using siRNA shuttled in exosomes [100]. In preclinical studies conducted in mice, the treatment reduced KRAS^D12G^ mRNA levels, which resulted in suppressed cancer cell proliferation and an overall increased lifespan [100,101,102]. Another siRNA therapeutic, Atu027, targets the protein kinase N3 (PKN3), which is crucial for cell proliferation and migration. In liver cancer, the inhibition of PKN3 showed a significant reduction in cell growth and metastasis formation in non-human primates [103]. This was confirmed in a phase I study conducted in patients with advanced solid tumors who experienced either a stabilization of the disease or a complete or partial regression of metastases after treatment with Atu027 [104]. The siRNA therapeutic TKM-080301 influences the Polo-like kinase (PLK1), which has been shown to be overexpressed in hepatocellular carcinoma (HCC). The knockdown of PLK1 by TKM-080301 reduced cell proliferation in patients with HCC [105]. The drug STP705 is a two-component product containing siRNAs directed against transforming growth factor β (TGFβ) and cyclooxygenase-2 (COX-2). TGFβ released by cancer cells inhibits the function of T cells in the tumor microenvironment (TME), and COX-2 acts as a proliferative mediator. In a human cholangiocarcinoma xenograft tumor model in nude mice, STP705 treatment reduced the immune inhibitory effect of the cancer cells and consequently inhibited tumor growth [106]. Little information is available on the results of the phase I clinical study of STP705 (NCT04293679); however, a phase II clinical study is already ongoing (NCT04844983). The therapeutic agent siRNA-EphA2-DOPC (EPHARNA) targets the expression of the ephrin type-A receptor 2 (EphA2), which is overexpressed in human cancers, including breast, prostate, lung, ovarian, endometrial, and pancreatic carcinomas. It has been shown that this receptor is involved in proliferation, survival, migration, invasion, and angiogenesis and, thereby, is associated with adverse outcomes in cancer patients. In turn, the administration of siRNA against EphA2 shuttled in 1,2-Dioleoyl-sn-Glycero-3-Phosphatidylcholine (DOPC) nanoliposomal particles down-regulated the EphA2 expression and decreased tumor cell growth shown in a murine ovarian cancer model [107]. In 2017, preclinical mammalian safety studies were conducted with EPHARNA [108], followed by the start of a clinical study phase I (ongoing, NCT01591356). The latest clinical trial (NCT06424301) targets refractory retinoblastoma utilizing siRNA against nudix hydrolase 21 modulators (NUDT21). As central factor in alternative polyadenylation, NUDT21 induces 3′UTR shortening of mRNAs often observed in cancer cells to avoid miRNA repression of oncogenes. By inhibiting NUDT21, cancer cell proliferation is inhibited and apoptosis induced [109]. 

In contrast to siRNA, which usually has a clearly defined target, one miRNA can suppress several targets, and miRNAs are part of a complex regulatory network. While they are frequently utilized in clinical trials as prognostic factors or for disease monitoring, the list of trials using miRNA-based therapeutics is currently short. The first clinical trial with a miRNA-based therapeutic tested MRX34, which is a miR-34a mimic. miR-34a is a naturally occurring tumor suppressor that is lost or expressed at reduced levels in a broad range of tumor types. miR-34a is involved in the down-regulation of the expression of over 30 oncogenes, including *myelocytomatosis oncogene* (*MYC*), *platelet derived growth factor receptor α* (*PDGFR-α*), *cyclin-dependent kinase* (*CDK*) *4/6*, and *B-cell lymphoma 2* (*BCL2*) [110]. The first phase I study with MRX34 (NCT01829971) (Table 1) was terminated because of several drug-related severe adverse events, comprising systemic enterocolitis, colitis/pneumonitis, systemic inflammatory response syndrome, and cytokine release syndrome with respiratory failure, and hepatic failure, of which the latter four led to death. The following MRX34 study (NCT02862145) (Table 1) never recruited patients because of the severe side effects in the previous study [110]. Another approach is TargmiRs, which is a miR-16 mimic designed to counteract the loss of the miR-15 and miR-16 family miRNAs in cancer cells. In preclinical models, the expression of miR-16 was restored in malignant pleural mesothelioma tumor specimens as well as in xenograft-bearing nude mice, resulting in a dose-dependent tumor growth inhibition [111]. In the following clinical study phase I (NCT02369198), 1 out of 26 patients showed a partial response, and 15 patients showed stable disease. Regarding the side effects, five dose-limiting toxicities occurred, but no drug-related deaths [112]. The discussed siRNA and miRNA therapeutics targeting cancer cells are depicted in the figure in Section 4.

**Table 1 ijms-25-07275-t001:** Overview of clinical trials exploiting siRNA and miRNA therapeutics in cancer therapy.

	Name of the Product	Cellular Target	Disease	Phase and NCT	Recruiting Status	Source
siRNA	siG12D LODER	KRAS^G12D^	Pancreatic cancer	Phase II NCT01676259	Unknown status	[98,99]
	iExosomes	KRAS^G12D^	Metastatic pancreas cancer	Phase I NCT03608631	Active	[100,101,102]
	Atu027	Protein kinase N3	Advanced solid tumors	Phase I NCT00938574	Completed	[103,104]
	TKM-080301	PLK1	Primary or secondary liver cancer	Phase I NCT01437007	Completed	[105]
	STP705	COX-2 and TGFβ	Cutaneous squamous cell carcinoma (in situ)	Phase II NCT04844983	Completed	[106]
	siRNA-EphA2-DOPC	EphA2	Advanced recurrent solid tumors	Phase I NCT01591356	Recruiting	[107,108]
	-	NUDT21	Refractory retinoblastoma	Phase I NCT06424301	Not yet recruiting	[109]
miRNA	MRX34	miR-34 mimic	Pulmonary lymphangitic carcinomatosis, small cell lung cancer, lymphoma, melanoma, multiple myeloma, renal-cell carcinoma, non-small cell lung cancer	Phase I NCT01829971	Terminated	[110]
			Melanoma	Phase I/II NCT02862145	Withdrawn	[110]
	TargmiRs	miR16 mimic	Malignant pleural mesothelioma and NSCLC	Phase I NCT02369198	Completed	[111,112]

## 4. Exploiting the Immune System for Cancer Therapy

While all the above-mentioned RNA treatments are designed to manipulate cancer cells, the cells of the immune system could represent a potential alternative target (Figure 3). The term immune surveillance emphasizes that the host’s immune system is able to detect tumor cells and even has the capability to inhibit tumor growth [113,114,115]. However, the tumor counteracts the immune surveillance by various immune escape mechanisms. This antagonism is delineated in the concept of the three phases of immune editing: eradiation, equilibrium, and escape [113,115]. One escape mechanism is the development of a tumor-supportive, immune-suppressive TME as an ecological niche for the cancer cells, established via soluble factors such as cytokines, chemokines, and metabolites, as wells as the upregulation of inhibiting surface receptors by cancer cells (e.g., immune checkpoints) [114,116,117,118].

The TME consists not only of cancer cells and cancer-cell-derived factors but also of several non-tumoral cells, including various immune cells [119]. These are innate immune cells such as monocytes, macrophages, dendritic cells (DCs), and natural killer (NK) cells, as well as adaptive immune cells such as T cells and B cells [120,121,122]. However, it became evident that the presence of these immune cells is not sufficient for controlling the tumor since their altered phenotype within the TME acts in favor of the tumor. This includes the induction of immune suppressive cells, the functional inhibition of pro-inflammatory immune cells, and the attraction of tolerance-promoting immune cells [119,120]. Consequently, the tumor-mediated pro-tumoral immune suppression and the anti-tumoral immune effects are locked in a constant battle, with most immune cells being able to fight both sides. Today, it is widely accepted that the localization, number, and phenotype of the immune cells are decisive for the clinical outcome [123].

In this complex microcosm, pro-inflammatory immune cells such as M1-like macrophages, conventional dendritic cells type 1 (cDC1), and cytotoxic CD8^+^ T cells exert anti-tumoral functions and are associated with improved survival for cancer patients [124,125,126]. In contrast to that, anti-inflammatory immune cell phenotypes such as M2-like macrophages, the myeloid-derived suppressor cells (MDSC), mature DCs enriched in immunoregulatory genes (mreg-DCs), regulatory T cells (Treg), and B cells (Breg), as well as low numbers of cytotoxic CD8^+^ T cells and cDC1 have been linked to a poor survival prognosis [127,128]. Here, it must be emphasized that the antigen-presenting cells (APCs) that process and present tumor antigen and induce T cell activation, especially, play a central role in the decision for an anti- or pro-tumoral immune response. For efficient control of the tumor, different immune cells are required: those that identify abnormal host cells and those that exert effector functions for the elimination of these cells.

### 4.1. Rethinking Cancer Immunotherapy—The Potential of Small Non-Coding RNAs

Current cancer treatments include several immunotherapeutic approaches that include immune checkpoint inhibitors that block inhibitory receptors on immune effectors cells or their ligands and thus overcome functional impairment of anti-tumor immune reactions, adoptive cell transfer therapy with ex vivo expanded immune cells which can be transfected with chimeric antigen receptors (CARs) or T cell receptors, monoclonal antibodies, antibody–drug conjugates, treatment vaccines (such as the oncolytic virus T-VEC or monocyte-derived DCs) and small molecule drugs (reviewed by Liu et al. [129]). The main problems affecting all immunotherapies are immune-resistance mechanisms, including the initially low mutational burden of the tumor cells that results in the absence of tumor-specific antigens, adaptive changes of cancer cells causing induced therapy resistance, and an immune suppressive environment formed by infiltrating immune cells such as Tregs, MDSCs and tumor-associated macrophages (TAMs) [130,131,132]. sncRNAs in immunotherapy have two potential applications: they can serve as predictive biomarkers and as therapeutic targets. As predictive markers, specific miRNAs in the blood and tumor tissue of cancer patients can correlate with either positive or negative outcomes of CAR-T cell therapy, cancer vaccines, immune checkpoint inhibition, and small molecule treatment, as reviewed extensively by Alahdal et al. [133]. sncRNAs may serve as direct targets not only in tumor cells to address pro-tumoral alterations in their miRNA expression but also within immune cells in the TME to direct them toward an anti-tumoral phenotype [134,135]. The database RNA2immune lists today (accessed June 2024) over 3600 human miRNAs that are associated with cancer immunology, including cancer cells and adjacent immune cells [136]. Similar to immune cells, sncRNAs can act in favor or in detriment of the tumor and can participate in immune escape or immune response [135]. In general, therapeutic approaches targeting the myeloid cells within the TME, especially addressing the immune-suppressive phenotypes, are quite rare but could be beneficial in combination with the previously listed cancer therapies. The next section summarizes which myeloid cells are possible targets and how reprogramming on a molecular level could be achieved. The term ‘reprogramming’ in the following stands for shifting the balance within the TME by inhibiting immune suppressive signals and enhancing pro-inflammatory effects to recapture the immune cells’ own anti-tumoral functions.

### 4.2. Targeting Myeloid Cells to Rectify the Tumor Microenvironment

#### 4.2.1. Macrophages

According to Mills et al., activated macrophages are divided into two phenotypes called classically activated and pro-inflammatory (M1) or alternatively activated and anti-inflammatory (M2), based on their polarization. Today, this clear division is widely discussed and shifted to the assumption that polarized macrophages in vivo form a spectrum. Since their polarization occurs as a response to a multitude of environmental influences, a variety of phenotypes in between the full M1 or M2 polarization are observed [137,138,139]. Despite this gradient, the phenotypical division is still made into M1 and M2 macrophages, with M1 macrophages expressing the surface markers CD86, CD80, CD68, and major histocompatibility complex class II (MHC-II), and the intracellular markers inducible nitric oxide synthetase (iNOS), interferon regulating factor 5 (IRF5), and signal transducer and activator of transcription (STAT) 1. On the contrary, M2 macrophages express the surface markers CD206, CD163, CD209, and the intracellular markers arginase 1, IRF4, and STAT3, while showing low expression levels of M1 markers [140,141].

In cancer research, TAMs represent macrophages in the TME, which are the main infiltrating immune cell population [142,143]. Regarding their phenotype, TAMs were initially classified as M2-like macrophages following Mills nomenclature [144]. Recently, it became evident that TAMs form at least two subgroups within the TME that are differently polarized into M1-like or M2-like phenotypes harboring different functionalities [145]. In this context, M1-like TAMs have the potential to kill tumor cells via antibody-dependent cellular cytotoxicity and phagocytosis and can induce anti-tumoral responses of innate and adaptive immune cells. However, in most established tumors, the environmental conditions formed by the tumor enhance the number of M2-like TAMs. This contributes to cancer progression and metastasis formation by several mechanisms, including promotion of tumor cell proliferation and survival, angiogenesis, and suppression of innate and adaptive immune responses [145,146].

Since environmental conditions drive the polarization of macrophages, the signaling molecules present in the TME as well as the receptor repertoire and the activity of internal signaling pathways of the macrophages, are substantially involved in the expression of either M1- or M2-related genes [147]. Table 2 summarizes key signaling pathways for macrophage polarization. For M1 polarization, tumor necrosis factor α (TNFα), CD40 ligand (CD40L), Toll-like receptors (TLR)-ligands, and granulocyte/macrophage colony-stimulating factor (GM-CSF) signaling converge in the induction of nuclear factor-κB (NFκB) downstream signaling as well as mitogen-activated protein kinase (MAPK), and serine/threonine-protein kinase (AKT) signaling. Apart from that, M1 polarization is induced via interferon γ (IFNγ) signaling that activates the interferon regulating factor 9 (IRF9), resulting in the expression of interferon-stimulated genes (ISGs). Further, M1 polarization is induced by IL12p40 and IL12p35 binding to IL-12R-β1 and β2, respectively. This leads to the translocation of STAT4 into the nucleus, where it binds to the promotor region of IFNγ targeted genes [148,149]. Another M1 driver is Angiopoietin 1 (ANG-1), which binds to the Tek tyrosine kinase receptor (TIE2). Upon ligand binding, the receptor induces signaling via p38 and extracellular signal-regulated kinas 1/2 (ERK1/2) independent of AKT activation [150].

The M2 polarization, which is driven by IL-4, IL-13, IL-10, IL-6, or TGFβ, uses different downstream signaling pathways, including STAT3 and STAT6 or the human mothers against decapentaplegic homologs 2, 3, and 4 (SMAD2/3/4). Further, M2 polarization is induced by colony-stimulating factor 1 (CSF-1), also known as macrophage colony-stimulating factor (M-CSF). Upon binding to the receptor CSF-R1, various singling pathways are induced, including phosphatidylinositol-3 kinase (PI3K)/AKT, jun N-terminal kinase (JNK)/ERK1/2, and phospholipase C γ 2 (PLC-γ2), which promote survival, proliferation, and differentiation of the macrophages [151,152]. Similar to ANG-1, ANG-2 binds to TIE2 but does not induce phosphorylation, thereby ANG-2 acts as a competitive inhibitor of ANG-1. This results in the upregulation of M2-associated genes, including IL-10, CCL17, and pro-angiogenic genes [153]. Additionally, the signaling axis of CD47 and the signal regulatory protein α (SIRPα) contribute to M2 polarization. Upon binding of the ligand CD47 to the receptor SIRPα, the immunoreceptor tyrosine-based inhibitory motif (ITIM) of SIRPα binds the cytoplasmatic Src homology 2 domain phosphatases 1 and 2 (SHP1/2), which are involved in several signal transduction process. The inhibition of SHP1/2 by SIRPα diminishes PI3K, NFκB, MAPK, and STAT1 activation and suppresses M1 polarization [154,155]. The same mechanism is observed for leukocyte immunoglobulin-like receptor (LILRB2)-binding HLA class I. The ITIMs bind SHP1/2 and thereby down-regulate MAPK and NFκB signaling [156,157].

For therapeutic reprogramming, enhancement of M1-driving signaling pathways and inhibiting M2-driving signaling pathways is desirable. Here, a therapeutic approach with non-coding RNAs could support the expression of M1-related genes and inhibit the expression of proteins that drive an M2 phenotype.

**Table 2 ijms-25-07275-t002:** Overview of macrophage signaling pathways involved in cell polarization.

	Signaling Molecule	Receptor on Macrophages	Signaling Pathways	Source
Pro-inflammatory, M1-like	TNFα	TNFR1/2	NFκB and MAPK	[158]
	CD40L	CD40	NFκB and MAPK	[159]
	IFNγ	IFNR	IRF9	[160]
	TLR-ligands	TLR2/3/4/5/6	MAPK and NFκB	[161,162]
	GM-CSF	GM-CSF receptor	AKT and NFκB	[163]
	IL-12p40 IL12p35	IL-12R-β1 IL-12R-β2	STAT4	[148,149]
	ANG-1	TIE2	P38/ERK1/2	[150]
Anti-inflammatory, M2-like	IL-4	IL-4R	STAT6, PI3K/AKT	[164]
	IL-13	IL-13R	STAT6	[165]
	IL-6	IL-6R	STAT3, NFκB	[166]
	IL-10	IL-10R	STAT3	[167]
	TGFβ	TGFβ receptor	SMAD2/3/4	[168]
	CSF-1 (M-CSF)	CSF-R1	PI3K/AKT, JNK/ERK1/2, PLC-γ2	[151,152]
	ANG-2	TIE2	No signaling	[153]
	CD47	SIRPα	SHP1/2 fishing	[154,155]
	HLA class I	LILRB	SHP1/2 fishing	[156,157]

Besides targeting internal signaling pathways that affect macrophage polarization, M1 and M2 macrophages can be functionally altered by inhibiting immune regulatory surface markers or by changing their cytokine expression. M1 macrophages secrete a variety of pro-inflammatory cytokines, including IL-6, IL-8, TNF, IL-1β, IL-12, and IL-23, nitric oxide (NO), reactive oxygen species (ROS), and down-regulate the expression of IL-10 [141]. M2 macrophages express high quantities of the cytokines IL-10 and TGFβ, the chemokines CCL17, CCL18, and CCL22, as well as the growth factors CSF-1 and vascular endothelial growth factor (VEGF) [141]. The role of M2 cytokines as central drivers of the immune escape underlines the critical role of all cytokines in the TME as key mediators of an anti-inflammatory or pro-tumoral immune response of several immune cells, not only of TAMs. Consequently, siRNA specifically targeting mRNA of these cytokines or receptors in the TME is a promising therapeutic strategy (Figure 3).

#### 4.2.2. Dendritic Cells

DCs are divided into several subgroups depending on their ontogeny and phenotype, with emerging subgroups and a detectable heterogeneity within the subgroups. Consequently, this field is continuously developing, as reviewed by Segura et al. [169]. However, one main distinction is made between plasmacytoid DCs and conventional DCs. The latter are further divided into cDC1 and cDC2 [169]. This review focuses specifically on the DC subsets cDC1 and cDC2 that harbor different activating functions within the immune system. cDC1s, on the one hand, mediate immune responses against intracellular pathogens, drive T helper cell (Th) 1 activation, and are crucial for an anti-tumor immune response. cDC2, on the other hand, are central in extracellular pathogen defense by driving Th2 and Th17 responses and have a rather controversial role in anti-tumor immune responses [170]. Regarding ontogeny, both subsets derive from precursor cells from the bone marrow which are already partially committed to become cDC1 or cDC2 [171]. The underlying parameters of cDC differentiation are poorly understood. However, it is believed that at least a part of pre-cDCs differentiate within the tissue depending on environmental signals [172]. Within the tumor tissue, reduced chemokine-mediated recruitment of cDC1s [125], as well as dysfunctional cDCs have been observed [128,173].

In line with that, immature DCs (named iDCs) were identified in tumor tissue that was unable to stimulate T cells [174] and additionally expressed low levels of the co-stimulatory molecules CD80 and CD86, or none at all [175,176]. Ex vivo stimulation experiments of these iDCs failed since CD86 and CD80 were not inducible upon exogenous stimulation with TNF or CD40L [177]. This implies that these iDCs are not merely unstimulated but are functionally inhibited and have lost the ability to react adequately upon stimulation. The presentation of tumor antigens by these iDC in the absence of co-stimulatory molecules misleads T cell responses and induces peripheral T cell tolerance and anergy, which supports the tumor in immune evasion [177,178,179].

When examining mature DCs within the TME of human and mouse non-small cell lung cancer (NSCLC) samples, Maier et al. identified mregDCs that expressed an altered set of maturation genes. This regulatory cell program was identified in cDC1s and cDC2s upon tumor antigen uptake [128], indicating that DCs are altered by tumor cells within the TME. This alteration of DCs represents a mechanism that can potentially be targeted by cell reprogramming in new therapeutic approaches.

As described above, DCs are the central antigen-presenting cells that induce T cell responses via direct cell-to-cell contact. Hence, the surface molecule expression regarding co-stimulatory molecules and immune regulatory molecules is of vital importance. Table 3 subsumes tolerance-promoting signaling pathways that can lead to a down-regulation of DC-mediated CD8^+^ T cell responses or induce Treg generation [180]. One main tolerance-promoting effect on DCs is the down-regulation of CD80/86 as well as CD40. It is well known that this can be mediated via prostaglandin E_2_ (PGE_2_) and VEGF derived from tumor cells or other anti-inflammatory immune cells within the TME to suppress the maturation and function of DCs [181,182,183]. Using RNAi mechanisms to block the expression of the main receptors involved in the inhibition of DC functions, such as VEGF and PGE_2_ receptors, could be exploited as a therapeutic approach. In addition to that, non-coding RNA could be used to rescue the expression of co-stimulatory molecules and help to avoid the induction of T cell tolerance. To this end, therapies based on non-coding RNAs could target the negative regulators (e.g., miRNAs) of the mRNAs encoding these co-stimulatory molecules. However, since there is a variety of inhibitory signals that act simultaneously on DCs, the overall success of these strategies is questionable and needs to be tested. Alternatively, the overexpression of miR181-d could be a useful therapeutic tool since it has been shown that the overexpression of this miRNA alone is sufficient to induce DC maturation accompanied by the upregulation of CD80 and CD83 [184]. Su et al. suggested that, as a mechanism of action, miR-181-d targets a negative regulator of the NFκB signaling pathway, resulting in constitutive activation of the NFκB signaling pathway and cell maturation [184]. The direct induction of co-stimulatory molecule expression via NFκB signaling is also achievable via TLR ligands or by mRNA-mediated expression of constitutively active members of the NFκB-signaling cascade [185], but it has to be tested, which method is more useful.

Another promising target of RNAi-based therapy is checkpoint molecules on the DC surface, such as programmed cell death ligand 1 (PD-L1) and T cell immunoglobulin mucin receptor 3 (TIM3). TIM3 functions as a negative regulator of innate immune responses against nucleic acid in an HMGB1-dependent mechanism. The abrogated response of DCs towards nucleic acids diminishes the therapeutic efficacy of chemotherapy and DNA vaccination in cancer therapy [186]. Additionally, the expression of additional checkpoint molecules, including lymphocyte activation gene 3 protein (LAG3), CD200R, inducible T cell co-stimulator and its ligand (ICOS(L)), and glucocorticoid-induced tumor necrosis factor receptor related protein (GITR), have been reported for cDC1s [187]. For therapeutic use, the down-regulation of PD-L1 and other inhibitory receptors is immensely important, and, therefore, siRNA-mediated phenotypical knockdowns would be useful (Figure 3).

Cytokines that inhibit DC function are the anti-inflammatory cytokines IL-10 and TGFβ. Of note, these cytokines also drive M2 macrophage polarization (Table 3) [188,189,190]. Consequently, inhibiting the expression of the referring cytokine receptors on both immune cell populations could change the functionality of these cells within the TME tremendously. Further, Kudo-Saito et al. reported that tumor cell-derived CCL2 also promotes regulatory DCs, which in turn induce immunosuppressive CD4^+^FOXP3^+^ regulatory T cells [191]. Furthermore, Maier et al. demonstrated that mregDCs upregulate the IL-4R and that the IL-4 signaling down-regulates the IL-12 production by DCs, which is a central cytokine in T cell stimulation. The IL-12 production could be reinduced by a blockage of IL-4 [128]. Besides IL-4 blocking antibodies, RNAi that targets the downregulation of IL-4R or the upregulation of IL-12 could prevent DCs from acquiring a regulatory phenotype and, in turn, increase the stimulation of cytotoxic CD8^+^ T cells instead of regulatory T cells, but this needs to be tested.

**Table 3 ijms-25-07275-t003:** Signaling pathways inhibiting dendritic cell functions in the TME.

Signaling Molecule	DC Marker Expression	Source
PGE_2_	MHCII↓ ^1^, CD40↓, PD-L1↑ ^2^	[181]
VEGF	MHCII↓, CD40↓, CD86↓, IL-12↓	[182,183]
HMGB1	TIM3↑	[186,192]
CCL2	HLA-DR↓, PD-L1↑	[191]
IL-10	CD86↓, CD80↓, CD40↓, CD83↓, PD-L1↑	[188,190]
TGFβ	MHCII↓, CD40↓, PD-L1↑ CD86↓, CD80↓, CD83↓, IL-12↓	[173,192,193]
IL-4	IL-12↓	[128]

^1^ ↓ downregulation, ^2^ ↑ upregulation.

#### 4.2.3. Myeloid-Derived Suppressor Cells

In general, MDSCs are the subject of a controversial discussion. Originally, it was thought that MDSCs are migrated myeloid cells that are diverted within the TME toward an immature, non-specific immunosuppressive phenotype [194]. Recently, this shifted to the assumption of pathologically activated polymorphonuclear (PMN-MDSCs) and monocytic (M-MDSCs) MDSCs, both harboring immune suppressive functions [195,196]. The accumulation of MDSCs is induced by various factors in the TME, including cyclooxygenase 2, prostaglandins, GM-CSF, M-CSF, stem cell factor (SCF), VEGF, IL-1β, IL-6, IL-4, IL-10, and TGFβ [197,198]. However, it is still difficult to discriminate these cells from healthy cells by surface marker expression [199]. So far, human MDSCs were described as HLA-DR^low^, CD33^+^, and CD11b^+^ while being negative for DC markers (except CD11b), as well as NK-cell and T cell markers [199]. Alternatively, Bronte et al. define PMN-MDSCs as CD11b^+^CD14^−^CD15^+^ or CD11b^+^CD14^−^CD66b^+^, and M-MDSCs as CD11b^+^CD14^+^HLA-DR^−/low^CD15^−^, with CD33 that can be used as a marker instead of CD11b [195]. Besides a characteristic surface marker expression, MDSCs showed changes in transcriptomic and metabolomic levels compared to normal polymorphonuclear cells, as well as the ability to inhibit other immune cells, including T cells, DCs, and NK cells [199]. In summary, MDSCs remain an interesting target within the TME, but since there is neither a clear definition nor a defined phenotype of these cells, there is not yet enough information available to target these cells. Hence, the current focus lies on macrophages and DCs as targets for reprogramming.

**Figure 3 ijms-25-07275-f003:**
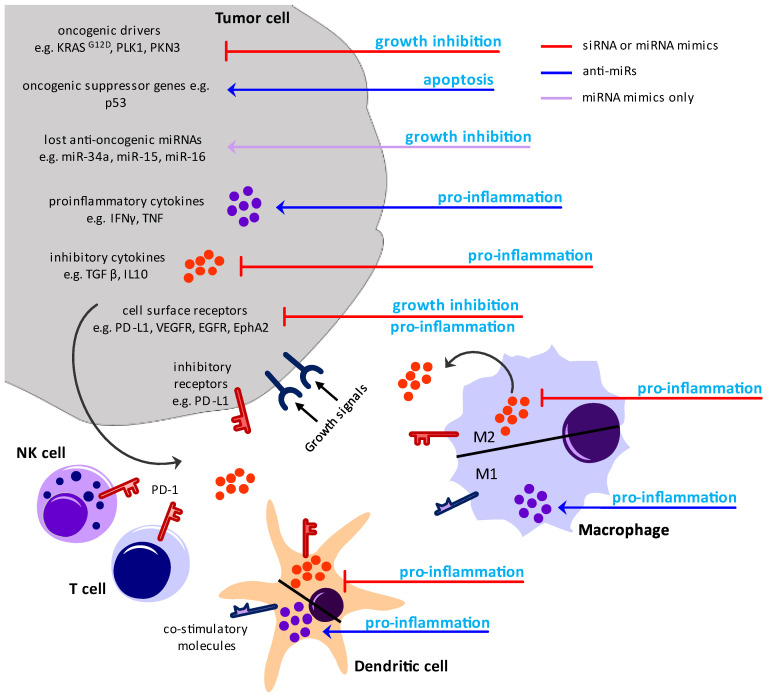
Therapeutical potential of small-interfering RNA and micro-RNA in cancer therapy. The therapeutical options are divided into two mechanisms of action: red hammers indicate the inhibition of pro-tumoral effects via small-interfering RNA (siRNA) or micro-RNA (miRNA) mimics of endogenous inhibitory miRNAs that knock down the referring mRNAs, blue arrows indicate the activation of anti-tumoral effects via anti-miRs that target miRNAs that are crucial in the negative regulation of these effects. An exception is the substitution therapy with miRNA mimics (violet arrow). For tumor cells, the inhibition of oncogenic drivers, immune suppressive cytokines, and pro-tumoral cell surface receptors are potential targets as well as the re-activation of oncogenic suppressor genes and pro-inflammatory cytokines. Thereby, the tumor growth is inhibited, cancer cell apoptosis is induced, and the suppressive microenvironment is shifted towards inflammation. For APCs like macrophages and dendritic cells, the inhibition of inhibitory cytokines and surface receptors (e.g., programmed cell-death ligand 1 (PD-L1)), as well as the activation of pro-inflammatory cytokines and the upregulation of co-stimulatory molecules, are therapeutical approaches to enhance inflammation and immune cell activation within the tumor microenvironment (TME). The reprogramming of inhibitory surface receptors and soluble factors of antigen-presenting cells (APCs) and cancer cells also affects other immune cells within the TME (e.g., natural killer (NK) cells and T cells), for example, via abrogating PD-L1/PD-1 interactions.

### 4.3. Reprogramming Experiments Targeting Myeloid Cells

While immunotherapies have largely focused on increasing the functionality of cytotoxic T cells (e.g., immune checkpoint blockade or CAR-T cells), treatments that intent to reprogram immune cells from a pro-tumoral to a pro-inflammatory phenotype with the usage of non-coding RNAs as a therapeutic tool have not been studied intensively. Nevertheless, there has been some respective research evaluating RNAi as immunotherapy for cancer, which will be summarized in the following.

Table 4 summarizes therapeutical approaches to reprogramming myeloid cells in murine cancer models. In a colon cancer model, Rossowska et al. down-regulated the expression of IL-10 in DCs and tumor cells with short-hairpin RNA (shRNA) shuttled via a lentiviral vector. This resulted in decreased IL-10 concentration and enhanced DC activity in the TME. Furthermore, reduced numbers of MDSCs and Tregs within the TME, an enhanced Th1-type anti-tumoral immune response, and delayed tumor growth were observed [200]. Cubillos-Ruiz et al. used a murine ovarian cancer model to demonstrate that anti-PD-L1 siRNA, shuttled via nanoparticles, was able to reprogram tumor-adjacent regulatory DCs towards activated DCs in vivo. The re-activated immune response resulted in an increased survival span of the mice compared to anti-PD-L1 antibody treatment [201].

Focusing on the reprogramming of macrophages, the repolarization of TAMs with M2-phenotype was investigated in a murine 4T1-based metastatic breast cancer model. Here, CpG-DNA as a TLR-agonist combined with an anti-IL-10R antibody shifted M2 into M1 polarized macrophages. The treatment was associated with inhibited metastasis formation and prolonged survival compared to the control group [202]. Also targeting TAMs, Zhang et al. demonstrated in an ovarian cancer model that in vivo reprogramming of M2-like macrophages by enhancing the NFκB signaling was achieved. In this approach, mRNA encoding interferon regulating factor 5 and the kinase IKKβ, both components of the NFκB signaling pathway, were applied simultaneously and packaged in nanoparticles. This treatment induced anti-tumoral immunity and tumor regression [203]. Mimicking the upregulating effects of these mRNAs enhancing the NFκB signaling, siRNA targeting negative regulators of the NFκB pathway could induce more stable M2-like macrophages, similar to miR-181-d. In another approach, a prostate cancer model and nanoparticles equipped with β-Cyclodextrin were used to target specifically M2 macrophages. As a therapeutic agent, siRNA directed against CSF-R1, which was utilized to abrogate the CSF-R1 signaling, a main driver of M2 polarization within the TME. Upon treatment, M1-related cytokines increased while M2 macrophage numbers decreased. Further, an increased cytotoxic T cell infiltration and inhibited tumor cell growth was observed [204]. Shobaki et al. utilized a mouse model with xenografts of human renal cancer and lipid nanoparticles to specifically shuttle siRNA targeting STAT3 and hypoxia-inducible factor 1-alpha (HIF-1α) to TAMs. The treatment resulted in increased numbers of macrophages and a higher abundance of the M1 polarized cells within the TME. Additionally, a decrease in angiogenesis and tumor growth was observed [205]. Furthermore, Qian et al. used a nanoparticle-based delivery system to target M2 macrophages within a murine 4T1 breast cancer model. Here, the therapeutic agent consisted of a CpG oligonucleotide and a siRNA directed against PI3Kγ (phosphoinositid-3-kinase γ). The conjugate molecule was designed to be cleaved within the cells by Dicer, so upon cleavage, the CpG oligonucleotide activates TLR9 while the siRNA mediates the inhibition of PI3Kγ and induces proinflammatory NFκB signaling. The researchers observed repolarization of M2 to M1 macrophages, tumor growth inhibition, and prolonged survival of the mice [206].

**Table 4 ijms-25-07275-t004:** Therapeutic approaches reprogramming myeloid cells in murine cancer models.

Model	Therapeutic	Target	Effect	Source
Colon cancer	shRNA	IL-10	IL-10↓ ^1^, DC activity↑ ^2^, Th1↑, MDSCs↓, Treg↓ Tumor growth↓	[200]
Ovarian cancer	siRNA	PD-L1	regDCs re-activated, survival↑	[201]
Breast cancer	CpG DNA + anti-IL-10R antibody	TLR, IL-10R	M2 macrophages repolarized to M1, metastasis formation↓, survival↑	[202]
Ovarian cancer	IRF5 and IKKβ mRNA	NFκB signaling	Anti-tumoral immunity↑, tumor growth↓	[203]
Prostate cancer	siRNA	CSF-1R	M1 cytokines↑, M2 numbers↓, cytotoxic T cell infiltration↑, tumor growth↓	[204]
Human xenograft in nude mice	siRNA	STAT3, HIF1-α	Macrophage infiltration in TME↑, M1↑, angiogenesis↓ tumor growth↓	[205]
Breast cancer xenograft	CpG + siRNA	TLR, PI3Kγ	M2 macrophages repolarized to M1, tumor growth↓, survival↑	[206]

^1^ ↓ downregulation, ^2^ ↑ upregulation.

In 2014, Troegeler et al. demonstrated efficient siRNA-mediated gene silencing in ex vivo-generated primary human monocytes, DCs, and macrophages [207]. Table 5 summarizes the following therapeutical approaches in myeloid cells in humans. Cheng et al. showed in isolated MDSCs from renal cancer patients’ blood that these cells were reprogrammed via intervening with the Notch signaling using siRNA. They previously observed that down-regulated Notch signaling in myeloid cells within tumor tissue drives an abnormal cell differentiation of these myeloid cells. Here, the Notch signaling was down-regulated by the kinase CK2 that phosphorylates Notch. Consequently, they used a siRNA directed against CK2, leading to reinitiation of the Notch signaling and, thereby, to normal cell differentiation [208].

Further, a clinical phase Ib trial (NCT03087591) applied siRNA therapy for solid tumors that could not be removed by surgery. Here, peripheral blood mononuclear cells (PBMCs) from the patients were collected via leukapheresis and ex vivo electroporated with siRNA that specifically knocks down Cbl proto-oncogene B (Cbl-b). This product, named APN401, was intravenously infused back into the patient. This resulted in increased production of IFNγ and IL-2 by the PBMCs in response to stimulation with anti-CD3/28, which implies that the function of T cells included in the PBMCs was central in this study [209]. Lastly, the clinical trial NCT04995536 aimed to evaluate the effect of the product named CAS3/SS3, which is a conjugate molecule of a CpG oligonucleotide and a siRNA directed against STAT3. With CAS3/SS3, the growth of tumor cells should be inhibited and immune cells activated. However, the study was withdrawn before starting the patient recruitment [210,211]. Currently, no further siRNA or miRNA therapy approaches targeting specifically myeloid cells in humans were found in preparation for this review.

**Table 5 ijms-25-07275-t005:** Therapeutic approaches reprogramming myeloid cells in humans.

Model	Therapeutic	Target	Effect	Source
Ex vivo MDSCs from renal cancer patient’s blood	siRNA	CK2 in NOTCH signaling	MDSCs reprogramming	[208]
Ex vivo PBMCs from solid tumor patients NCT03087591	siRNA (APN401)	Cbl-b	IFNγ↑ ^1^, IL2↑ production of PBMCs	[209]
In vivo NCT04995536 (withdrawn)	CpG + anti-STAT3 siRNA (CAS3/SS3)	TLR9, STAT3	-	[210,211]

^1^ ↑ upregulation.

## 5. Conclusions

Myeloid cells within the TME represent a double-edged sword in anti-tumor immune responses. Since these cells harbor a central role in directing and regulating tumor-specific immune reactions and are present in the TME in high numbers, proper pro-inflammatory functionality of these cells is crucial for a strong anti-tumoral immune response. Re-directing of the immune response can be achieved by reprogramming these myeloid cells in vivo by exploiting siRNA- or miRNA-based therapeutics, as already shown in mice. However, further research regarding driver signaling pathways and complex signaling networks is necessary.

## Figures and Tables

**Figure 2 ijms-25-07275-f002:**
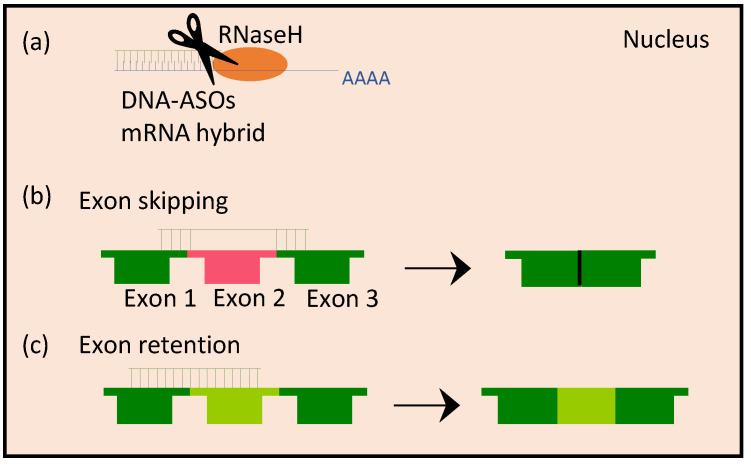
Overview of mechanisms of actions of anti-sense oligos (ASOs) within the nucleus. (**a**) DNA ASOs bind to mRNA within the nucleus to form DNA-RNA hybrids, which are recognized and cleaved by the enzyme RNaseH. (**b**) ASO can mediate exon skipping by jointing two exons that are not adjacent. (**c**) ASOs can mediate exon retention by binding to adjacent exons and thereby retaining the exon within the mRNA processing.

## Data Availability

Not applicable.

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
