# Peer review of "Non-Coding RNA in Tumor Cells and Tumor-Associated Myeloid Cells—Function and Therapeutic Potential"

_ijms, 2024, doi:10.3390/ijms25137275_

Round 1

Reviewer 1 Report

Comments and Suggestions for Authors

Review

Authors endeavoured to summarize in a review the current state-of-the-art for less-known RNA types such as small interfering (si)RNA and micro (mi)RNA. Also, they concisely summarised various RNA-based technologies, corresponding therapeutic applications and possibilities, particularly targeting tumors. The manipulation of tumor-associated immune cells and especially tumor-associated myeloid cells are in details discussed.

The review starting with the short general overview and few historical facts, progresses  systematically into more focused domains. It embraces 186 references, mostly covering the last 15 years. 

However, comparative data are presented in only 3 Tables, with very few columns and rows, comparison of novel data in last decades is presented on 7 pages and only two figures, which actually present well-known mechanisms of action.  One would expect that from almost 200 references more diagrams/Schemes and comparative Tables would be composed.

Thus, I would recommend more graphical and/or table presentations to help reader more efficiently in collection of referenced litarture.

Author Response

Reviewer 1

First of all, we thank the reviewer for reading and evaluating our manuscript. To the raised concerns, we have prepared a point-by-point response below:

Please note that all page and line numbers indicated refer to the document version with the track changes shown inline.

Authors endeavoured to summarize in a review the current state-of-the-art for less-known RNA types such as small interfering (si)RNA and micro (mi)RNA. Also, they concisely summarised various RNA-based technologies, corresponding therapeutic applications and possibilities, particularly targeting tumors. The manipulation of tumor-associated immune cells and especially tumor-associated myeloid cells are in details discussed.

The review starting with the short general overview and few historical facts, progresses  systematically into more focused domains. It embraces 186 references, mostly covering the last 15 years.

However, comparative data are presented in only 3 Tables, with very few columns and rows, comparison of novel data in last decades is presented on 7 pages and only two figures, which actually present well-known mechanisms of action. One would expect that from almost 200 references more diagrams/Schemes and comparative Tables would be composed. Thus, I would recommend more graphical and/or table presentations to help reader more efficiently in collection of referenced literature.

Answer: We have received this comment and extended table 1 (page 1, 371) by including two more clinical trials. Now, all clinical trials for miRNA and siRNA therapeutics listed at Clincialtrials.gov are included. Additionally, we extended table 2 (page 13, line 533) by adding more relevant signaling pathways for macrophage polarization and added a column for the references to have easily access to further information. Further, we added 2 more tables: table 4 (page 18, line 729) and 5 (page 19, line 755) for a more comparative overview of all reprogramming experiments in mice and humans discussed in this review. Furthermore, a third figure (page 16, line 661) as graphical summary of the therapeutic potential in cancer cells and myeloid cells within the TME was added.

Reviewer 2 Report

Comments and Suggestions for Authors

In this manuscript, Binder et. al. have explored and documented an exciting avenue of the role of non-coding RNA in tumor microenvironment, with emphasis on immune cells and their therapeutic application. This is an interesting area but there are There are several areas that need improvement:

  1. The authors need to make sure that the writing, English, and flow of the manuscript are as per the professional scientific journals.
  2. For instance, the sentence "Outstanding is the IFNγ signaling which induces the expression of 408 interferon stimulated genes (ISGs) via the interferon regulating factor 9 (IRF9). T"
  3. The authors can replace "medical potential" with therapeutic potential or any other suitable wording.
  4. The authors need to rewrite the abstract for clarity and should start with an introductory sentence related to ncRNA and then expand on its role. The authors can exclude "grand finale" and "Nobel prize".
  5. The abstract should crystallize the whole manuscript, and attract readers to read the article. The authors can restructure the abstract to reflect the same.
  6. The authors need to add an introduction section.
  7. "The authors can exclude first references, such as "Ours".
  8. The authors need to expand a segment on immunotherapy and the role of non-coding RNA. Broadly, they can start with the introduction of immune system and gradually plug in the role of ncRNA into it. The authors can also add a figure to illustrate "the medical" or "the therapeutic" potential through the figure. It would greatly improve the reach of the manuscript.
Comments on the Quality of English Language

Extensive editing of the English language is required.

Author Response

Reviewer 2

First of all, we thank the reviewer for reading and evaluating our manuscript. To the raised concerns, we have prepared a point-by-point response below:

Please note that all page and line numbers indicated refer to the document version with the track changes shown inline.

  1. English language

  • The authors need to make sure that the writing, English, and flow of the manuscript are as per the professional scientific journals.
  • For instance, the sentence "Outstanding is the IFNγ signaling which induces the expression of 408 interferon stimulated genes (ISGs) via the interferon regulating factor 9 (IRF9). T"

Answer: The English language was edited and corrected throughout the whole manuscript, including the respective sentence (page 12, lines 500-502).

  1. The authors can replace "medical potential" with therapeutic potential or any other suitable wording.

Answer: We adjusted the headline to therapeutic potential (page 1, line 3).

  1. Clarification of the abstract

  • The authors need to rewrite the abstract for clarity and should start with an introductory sentence related to ncRNA and then expand on its role. The authors can exclude "grand finale" and "Nobel prize".
  • The abstract should crystallize the whole manuscript, and attract readers to read the article. The authors can restructure the abstract to reflect the same.

Answer: We appreciated this comment and completely rewrote the abstract (page 1 lines 12-24).

  1. The authors need to add an introduction section.

Answer: We added an introduction section (page 1 and 2, lines 28-60).

  1. "The authors can exclude first references, such as "Ours".

Answer: We excluded all first forms throughout the whole manuscript.

  1. The authors need to expand a segment on immunotherapy and the role of non-coding RNA. Broadly, they can start with the introduction of immune system and gradually plug in the role of ncRNA into it.

Answer: We expanded the segment on immunotherapy. Therefore, we restructured the manuscript and extended section 4 to:  ‘4. Exploiting the immune system for cancer therapy’ (page 10, lines 374-413) There we start with discussing the role of the immune system in cancer therapy. Additionally, we added the section 4.1. Re-thinking cancer immunotherapy – the potential of sncRNAs’ (page 11, lines 415-450) to give a short summary of current immunotherapy approaches and to outline the potential of sncRNAs in cancer therapy.

  1. The authors can also add a figure to illustrate "the medical" or "the therapeutic" potential through the figure. It would greatly improve the reach of the manuscript.

Answer: We added a third figure (page 16, line 661) as overview of the therapeutic potential of siRNA and miRNA therapeutics in cancer therapy to give a graphical summary.
